# *Legionella pneumophila* modulates host energy metabolism by ADP-ribosylation of ADP/ATP translocases

Jiaqi Fu[1,2], Mowei Zhou[3], Marina A Gritsenko[4], Ernesto S Nakayasu[4], Lei Song[1]*, Zhao-Qing Luo[2]*

[1]Department of Respiratory Medicine and Center of Pathogen Biology and Infectious Diseases, Key Laboratory of Organ Regeneration and Transplantation of the Ministry of Education, State Key Laboratory of Zoonotic Diseases, The First Hospital of Jilin University, Changchun, China; [2]Department of Biological Science, Purdue University, West Lafayette, United States; [3]Environmental and Molecular Sciences Division, Pacific Northwest National Laboratory, Richland, United States; [4]Biological Science Division, Pacific Northwest National Laboratory, Richland, United States

*For correspondence:
lsong@jlu.edu.cn (LS);
luoz@purdue.edu (Z-QingL)

**Competing interest:** The authors declare that no competing interests exist.

**Abstract** The intracellular pathogen *Legionella pneumophila* delivers more than 330 effectors into host cells by its Dot/Icm secretion system. Those effectors direct the biogenesis of the *Legionella*-containing vacuole (LCV) that permits its intracellular survival and replication. It has long been documented that the LCV is associated with mitochondria and a number of Dot/Icm effectors have been shown to target to this organelle. Yet, the biochemical function and host cell target of most of these effectors remain unknown. Here, we found that the Dot/Icm substrate Ceg3 (Lpg0080) is a mono-ADP-ribosyltransferase that localizes to the mitochondria in host cells where it attacks ADP/ATP translocases by ADP-ribosylation, and blunts their ADP/ATP exchange activity. The modification occurs on the second arginine residue in the -RRRMMM- element, which is conserved among all known ADP/ATP carriers from different organisms. Our results reveal modulation of host energy metabolism as a virulence mechanism for *L. pneumophila*.

## Editor's evaluation

In this paper you show that the intracellular pathogen Legionella pneumophila uses an ADP-ribosyltransferase, Ceg3, to target mitochondria in infected cells. Ceg3 interferes with ADP/ATP exchange thereby modulating the energy metabolism in infected cells. This provides interesting new insight into the manipulation of mitochondrial function by an intracellular bacterium.

## Introduction

*Legionella pneumophila,* a Gram-negative intracellular bacterial pathogen, is the causative agent of Legionnaires' disease. This bacterium exists ubiquitously in the environment as a parasite of freshwater amoebae (*Richards et al., 2013*). Infection of humans occurs when susceptible individuals inhale aerosol contaminated by bacteria and introduce the pathogen to the lungs where it is phagocytosed by alveolar macrophages. Instead of being digested, engulfed bacteria survive and replicate in macrophages, leading to tissue damage and strong inflammatory responses, and the development of disease symptoms (*Newton et al., 2010*). The cell biological characteristics of infected amoebae and mammalian cells are highly similar, both are featured by the formation of an endoplasmic reticulum (ER)-like phagosome called the *Legionella*-containing vacuole (LCV), which in the initial phase of

infection, bypasses the endocytic pathways of the phagocytes (*Isberg et al., 2009*). The biogenesis of the LCV requires the Dot/Icm system, which is arguably the most important pathogenic factor of *L. pneumophila* (*Hubber and Roy, 2010*). Mutations in any component gene essential for the function of the Dot/Icm transporter result in a complete loss of virulence on all hosts. This multicomponent machine injects more than 330 virulence factors known as effectors into host cells to construct the LCV by manipulating diverse cellular processes, including vesicle trafficking, autophagy, lipid metabolism, and cytoskeleton structure via distinct biochemical activities (*Qiu and Luo, 2017*).

A common mechanism utilized by bacterial effectors is to function as enzymes that attack specific host proteins involved in important cellular processes by posttranslational modifications (PTMs) (*Salomon and Orth, 2013*), including phosphorylation (*Lee and Machner, 2018*), AMPylation (*Müller et al., 2010*), phosphorylcholination (*Mukherjee et al., 2011*; *Tan et al., 2011*), ubiquitination (*Gan et al., 2019*; *Qiu et al., 2016*), and ADP-ribosylation (*Qiu et al., 2016*; *Black et al., 2021*). Among these, ADP-ribosylation is one of the first identified PTMs utilized by toxins from bacterial pathogens (*Aravind et al., 2015*); this modification is catalyzed by ADP-ribosyltransferases (ARTs) that transfer the ADP-ribose moiety from nicotinamide adenine dinucleotide (NAD) to target substrates via an *N*-, *O*-, or *S*-glycosidic linkage (*Cohen and Chang, 2018*). Depending on the property of the enzymes, the modification can be the addition of either one (mono) or more (poly) ADP-ribosyl moieties onto the recipient site of the target proteins (*Hottiger et al., 2010*). In addition to virulence factors from bacterial pathogens, ARTs have been identified in eukaryotic proteins, which regulate such important cellular processes as DNA damage repair, gene expression, and aging (*Palazzo et al., 2017*). Differing from ARTs of bacterial origins, which often are mono-ADP-ribosyltransferases (mARTs), many mammalian enzymes are poly (ADP-ribose) polymerases (PARPs) that induce the formation of poly ADP-ribose (PAR) chains on substrate molecules (*Hottiger et al., 2010*).

Bacterial toxin-induced mono-ADP-ribosylation attacks a wide spectrum of host functions to benefit the pathogens. Some of the best studied examples include the inhibition of host protein synthesis by the diphtheria toxin that modifies elongation factor 2 (EF2) (*Honjo et al., 1968*) and the interference of the host second messenger cAMP production by the cholera toxin, which ADP-ribosylates the Gαs subunit of the adenylate cyclase and locks the enzyme in an active form (*Cassel and Pfeuffer, 1978*). A number of virulence factors injected into host cells by specialized protein secretion systems also use ADP-ribosylation to modulate host functions. For example, the type III effector ExoS of *Pseudomonas aeruginosa* targets multiple cellular proteins, including Ras, the modification of which leads to the inhibition of ROS production by neutrophils (*Vareechon et al., 2017*). Recently, *Xu et al., 2019* found that the T3SS effector SopF from *Salmonella enterica* serovar Typhimurium attacks the ATP6V0C subunit of v-ATPase by ADP-ribosylation, thus blocking the recruitment of ATG16L1 to suppress autophagy (*Xu et al., 2019*). ADP-ribosylation of important signaling molecules such as ubiquitin has also been documented. One such example is CteC from *Chromobacterium violaceum*, which specifically modifies ubiquitin on Thr66 with a cryptic mART motif to disrupt host ubiquitin signaling (*Yan et al., 2020*).

Modification by mono-ADP-ribosylation has recently emerged as an important arsenal for *L. pneumophila* virulence. The effector Lpg0181 was found to inactivate the glutamate dehydrogenase in host cells using an mART activity (*Black et al., 2021*). Members of the SidE effector family catalyze ubiquitination of multiple host proteins by first activating ubiquitin via ADP-ribosylation on Arg42 with an mART activity (*Qiu et al., 2016*). The activated ubiquitin (ADPR-Ub) is then utilized by a phosphodiesterase activity embedded in the same proteins to catalyze the transfer of phosphoribosyl ubiquitin to serine residues of substrates (*Qiu et al., 2016*; *Bhogaraju et al., 2016*; *Kotewicz et al., 2017*).

Herein, we show that the *L. pneumophila* effector Ceg3 (Lpg0080) localizes to the mitochondrion where it targets carrier proteins of the ADP/ATP translocase family by ADP-ribosylation, leading to the inhibition of the ADP/ATP exchange in mitochondria. Our results add further weight to the modulation of energy metabolism as a virulence mechanism for a bacterial pathogen.

## Results

### The *L. pneumophila* effector Ceg3 is a putative mART that localizes to the mitochondrion

A number of *L. pneumophila* effectors have been found to use mART activity to modulate host functions ranging from metabolism to ubiquitination (*Qiu et al., 2016*; *Black et al., 2021*). To identify

additional effectors with potential mART activity, we used the HHpred algorithm (*Söding et al., 2005*) to analyze established Dot/Icm substrates (*Zhu et al., 2011*; *Burstein et al., 2016*). These efforts identified Ceg3 (Lpg0080), a 255-residue protein as a putative mART. The key residues in the predicted mART of Ceg3 are identical to those of bacterial toxins or effector proteins, including ExoS of *P. aeruginosa* (*Vareechon et al., 2017*), CtxA of *Vibrio cholera* (*Cassel and Pfeuffer, 1978*), and members of the SidE family of *L. pneumophila* (*Qiu et al., 2016*). The sequence of the predicted mART in Ceg3 is $R_{44}$-$S_{94}$-$E_{141}KE_{143}$, which resides in its central region (*Figure 1A*).

Ceg3 was first identified as a Dot/Icm substrate by virtue of being coregulated with known effector genes (*Zusman et al., 2007*) and was later shown to kill the yeast *Saccharomyces cerevisiae* upon ectopic expression (*Urbanus et al., 2016*). To determine whether the putative mART motif plays a role in its yeast toxicity, we constructed the $Ceg3_{E/A}$ mutant in which both of the predicted catalytic sites E141 and E143 were replaced with alanine. Compared to wild-type protein that severely inhibited yeast growth, $Ceg3_{E/A}$ has completely lost the toxicity (*Figure 1B*). These results suggest that the putative mART motif is critical for the activity of Ceg3.

To understand the function of Ceg3, we first determined its cellular localization. Results from pilot experiments indicated that the distribution of Flag-Ceg3 is not cytosolic. Further experiments using a mitotracker reagent revealed that Ceg3 colocalizes extensively with this mitochondrial marker with a Pearson's correlation coefficient value of 0.87 (*Figure 1C*), suggesting that Ceg3 localizes to the mitochondria. Similarly, immunogold labeling results using transmission electron microscopy showed that Flag-Ceg3 was associated with the mitochondria (*Figure 1D*). Thus, Ceg3 likely is targeted to the mitochondria.

We further analyzed the subcellular localization of Ceg3 by cell fractionation. Cells transfected to express Flag-Ceg3 or Flag-MavC were mechanically lysed and the mitochondrial fractions were obtained by centrifugation as described (*Chen et al., 2009*). Samples from each fraction were separated by SDS-PAGE and analyzed by immunoblotting with antibodies against Flag and resident proteins of the relevant organelles, respectively. Flag-Ceg3 co-fractionated with the subunit α1 of pyruvate dehydrogenase E1 (PDHA1), which is an established mitochondrial protein (*Børglum et al., 1997*). In agreement with results from immunostaining experiments, Flag-Ceg3 did not co-fractionate with Calnexin or GM130, markers for the ER and the Golgi apparatus. In contrast, Flag-MavC co-fractionated with the cytosolic protein tubulin (*Figure 1E*). These results corroborate well with the immunostaining data, further indicating mitochondrial localization of Ceg3. Finally, we detected Ceg3 in mitochondrial fractions of cells infected with a wild-type *L. pneumophila* strain overexpressing Flag-tagged Ceg3 but not with a mutant defective in the Dot/Icm transporter (*Figure 1F*), suggesting that Ceg3 is targeted to mitochondria once being injected into host cells.

The association of Ceg3 with mitochondria can be mediated either by binding to mitochondrial proteins or by integrating into its membranes. To distinguish between these two possibilities, we treated mitochondria isolated from cell expressing HA-Ceg3 with a high pH buffer (0.1 M $Na_2CO_3$, pH 11) and separated integral and peripheral membrane proteins by centrifugation. Two peripheral membrane proteins, cytochrome c (Cyto-C) and the beta subunit of ATP synthase (ATPB) that are associated with inner mitochondrial membranes (*Walker and Dickson, 2006*; *Garrido et al., 2006*), can be stripped from the mitochondria effectively by the high pH buffer (*Figure 1G*). In contrast, HA-Ceg3 was only detected in the pellet fraction together with the mitochondrial import receptor subunit TOM20 homolog (Tom20) and the voltage-dependent anion-selective channel protein 1 (VDAC1), two integral mitochondrial membrane proteins (*Burri et al., 2006*; *Hiller et al., 2008*). Thus, Ceg3 is not peripherally associated with mitochondrial membranes.

## ADP/ATP translocases (ANTs) are the cellular targets of Ceg3

To identify the cellular targets of Ceg3, we first determined its ability to induce ADP-ribosylation of host proteins upon transfection. To this end, we used a pan-ADP-ribose antibody (*Shin et al., 2020*) to detect ADP-ribosylated proteins in cell transfected to express Ceg3 or the $Ceg3_{E/A}$ mutant. Signals representing proteins of slightly larger than 25 kDa were detected in lysates of cells expressing Ceg3 but not in samples expressing $Ceg3_{E/A}$ (*Figure 2A*, left and middle panels). Probing these samples with antibodies specific for Flag and ADP-ribose simultaneously detected two proteins of distinct sizes in samples expressing Flag-Ceg3, indicating that the protein detected by the ADP-ribose antibody was

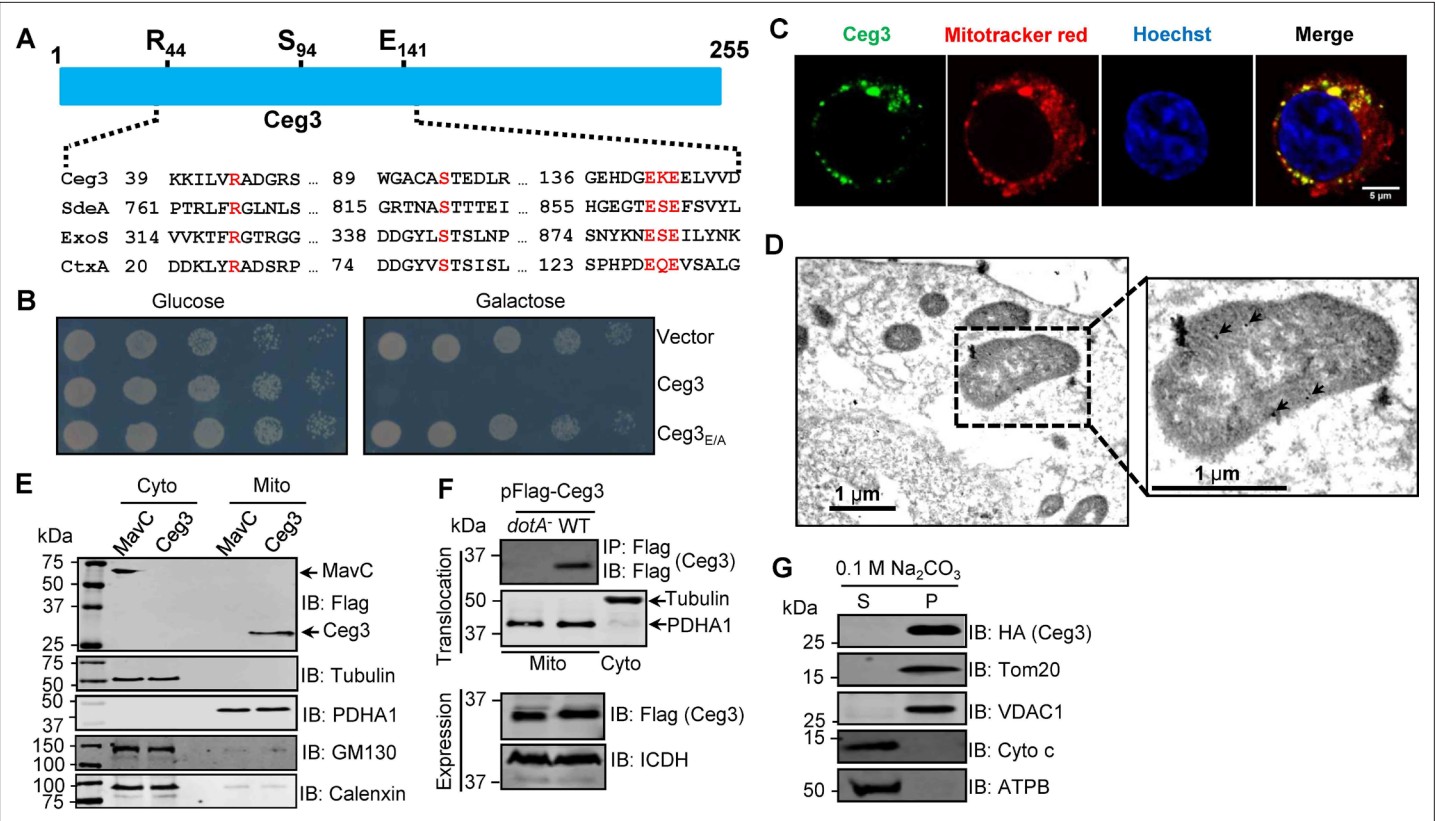

**Figure 1.** Ceg3 is a mitochondria-associated effector that inhibits yeast growth by a putative mART activity. (**A**) Sequence alignment of the central region of Ceg3 with three bacterial proteins of mART activity. The strictly conserved residues essential for catalysis are in red. SdeA, ExoS, and CtxA are from *Legionella pneumophila*, *Pseudomonas aeruginosa*, and *Vibrio cholerae*, respectively. (**B**) The predicted mART motif is critical for Ceg3-mediated yeast toxicity. Serially diluted yeast cells expressing Ceg3 or Ceg3$_{E/A}$ from a galactose-inducible promotor were spotted on the indicated media for 3 days before image acquisition. Similar results were obtained in at least three independent experiments. (**C**) Ceg3 co-localizes with a mitochondrial marker. HeLa cells were transfected with plasmids that direct the expression of Flag-Ceg3 for 18 hr and then incubated with fresh cell medium containing Mitotracker Red for 30 min. Cells were fixed and immunostained with antibodies specific for Flag (green). Images were acquired with a Zeiss LSM 880 confocal microscope. Scale bar, 5 µm. The colocalization of Ceg3 with mitochondria was quantitated by Pearson correlation coefficient with ImageJ. (**D**) Flag-Ceg3 localized to the mitochondria detected by Immunogold labeling. HEK293T cells expressing Flag-Ceg3 were fixed, stained with a mouse anti-Flag antibody and an anti-mouse IgG conjugated with 10 nm gold particles sequentially. Images were acquired with a Tecnai T12 electron microscopy. Area highlighted by rectangles (dashed line) on the left panel is magnified in the right panels. Gold particles were indicated by white arrows. Scale bar, 1 µm. (**E**) Ceg3 fractionated with mitochondria. The cytosol and mitochondrial fractions of HEK293T cells transfected to express Flag-Ceg3 or Flag-MavC were probed with antibodies specific for the indicated proteins. (**F**) Ceg3 translocated into host cells by the Dot/Icm system is targeted to mitochondria in cells infected with *L. pneumophila*. HEK293T cells expressing the FcγII receptor were infected with opsonized bacteria expressing 5xFlag-Ceg3 at an MOI of 100 for 2 hr. Lysates of isolated mitochondria were subjected to immunoprecipitation with beads coated with the Flag antibody, precipitates resolved by SDS-PAGE were detected by the Flag-specific antibody (top). The quality of cell fractionation was evaluated by detecting PDHA1 and tubulin. A cytosolic fraction sample was included as an additional control (middle). The expression of Ceg3 in bacteria was detected with the Flag antibody, and ICDH was probed as a loading control (bottom). (**G**) Ceg3 is an integral mitochondrial membrane protein. Mitochondria isolated from HEK293T cells expressing HA-Ceg3 were subjected to extraction with 0.1 M Na$_2$CO$_3$ (pH 11) for 30 min. Relevant proteins in soluble (S) and pellet (P) fractions separated by centrifugation were probed by immunoblotting with the indicated antibodies. MOI, multiplicity of infection.

not self-modified Flag-Ceg3 (***Figure 2A***, right panel). Thus, Ceg3 may ADP-ribosylate one or more host proteins with a size of approximately 25 kDa.

We also performed immunoprecipitation (IP) with beads coated with the Flag antibody in lysates from cells transfected to express Flag-Ceg3 or Flag-Ceg3$_{E/A}$. Several protein bands, including one that migrated at approximately 25 kDa were detected in samples transfected with Flag-Ceg3 or Flag-Ceg3$_{E/A}$ but not the empty vector (***Figure 2B***, right panel). Furthermore, when probed with the ADP-ribose antibody, strong signals were detected in samples expressing Flag-Ceg3 but not the mutant

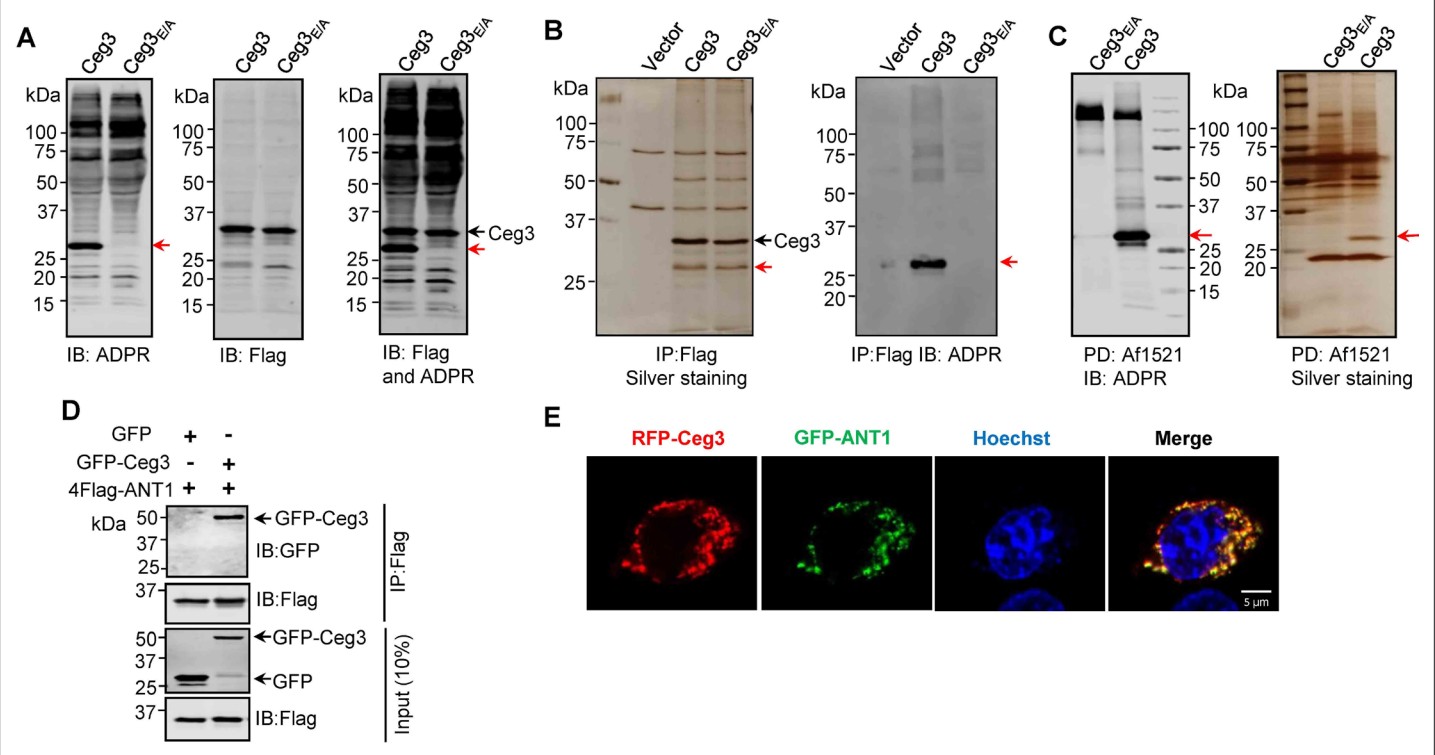

**Figure 2.** Identification of the cellular targets of Ceg3. (**A**) Detection of modified proteins by the ADPR-specific antibody. Lysates of HEK293T cells expressing 4xFlag-Ceg3 or 4xFlag-Ceg3$_{E/A}$ were probed with antibodies specific for ADPR-modification (left), the Flag tag (middle), or both (right). Note that the band indicated by a red arrow in samples expressing Ceg3 detected by the ADPR antibody represents its potential targets. (**B**) Substrate probing by immunoprecipitation (IP). Lysates of HEK293T cells transfected to express 4xFlag-Ceg3 or 4xFlag-Ceg3$_{E/A}$ were subjected to IP with beads coated with the Flag antibody and the products resolved by SDS-PAGE were detected by silver staining (left) or probed with the ADPR-specific antibody (right). Note the presence of a band in samples expressing Ceg3 but not Ceg3$_{E/A}$ when detected with the ADPR antibody (red arrows). (**C**) Enrichment of ADP-ribosylated proteins by Af1521 from cells transfected to express Ceg3. Lysates of HEK293T cells expressing 4xFlag-Ceg3 or 4xFlag-Ceg3$_{E/A}$ were incubated with beads coated with recombinant Af1521 and the pulldown products resolved by SDS-PAGE were detected by immunoblotting with the ADP-ribose antibody (left) or by silver staining (right) (red arrows). (**D**) Interactions between ANT1 and Ceg3. Lysates of HEK293T cells co-transfected to express 4xFlag-ANT1 and GFP-Ceg3 (or GFP) were subjected to IP with beads coated with the Flag antibody and bound proteins resolved by SDS-PAGE were detected with Flag and GFP antibodies, respectively. The expression of 4xFlag-ANT1, GFP-Ceg3, and GFP were similarly probed in total cell lysates as input. (**E**) Colocalization of Ceg3 and ANT1. HeLa cells transfected to express RFP-Ceg3 and GFP-ANT1 were fixed and analyzed. Images were acquired using a Zeiss LSM 880 confocal microscope. Scale bar, 5 μm. The colocalization of Ceg3 with ANT1 was quantitated by Pearson correlation coefficient with ImageJ.

with a disrupted mART motif (*Figure 2B*, left panel). Taken together, these results suggest that Ceg3 interacts with one or more host proteins of approximately 25 kDa.

To identify the substrates modified by Ceg3, we enriched ADP-ribosylated proteins from samples of cells transfected to express this effector by using recombinant Af1521, a protein from *Archaeoglobus fulgidus* that contains a macro domain involved in binding ADP-ribose moieties on modified proteins (*Allen et al., 2003*). After enrichment, the potential targets of Ceg3 with a molecular weight slightly higher than 25 kDa were detected by silver staining as well as ADP-ribose immunoblotting only in samples expressing Ceg3 (*Figure 2C*). This protein band detected by silver staining was then excised and analyzed by mass spectrometry. Among the top 10 proteins with the most hits, with the exception of keratin, a common contaminant in mass spectrometric samples, 4 were assigned as ADP/ATP translocases (ANTs) (*Supplementary file 1*). We confirmed ANT proteins as substrates of Ceg3 by transfecting mammalian cells to express GFP-Ceg3 and Flag-ANT1, which is one isoform of ADP/ATP translocases, and examined their interactions by IP. Precipitates obtained by beads coated with the Flag antibody contained GFP-Ceg3 (*Figure 2D*), indicating the binding between these two proteins. Consistent with these results, fluorescence signals of RFP-Ceg3 colocalize extensively with those of GFP-ANT1, with a Pearson's correlation coefficient value of 0.87 (*Figure 2E*).

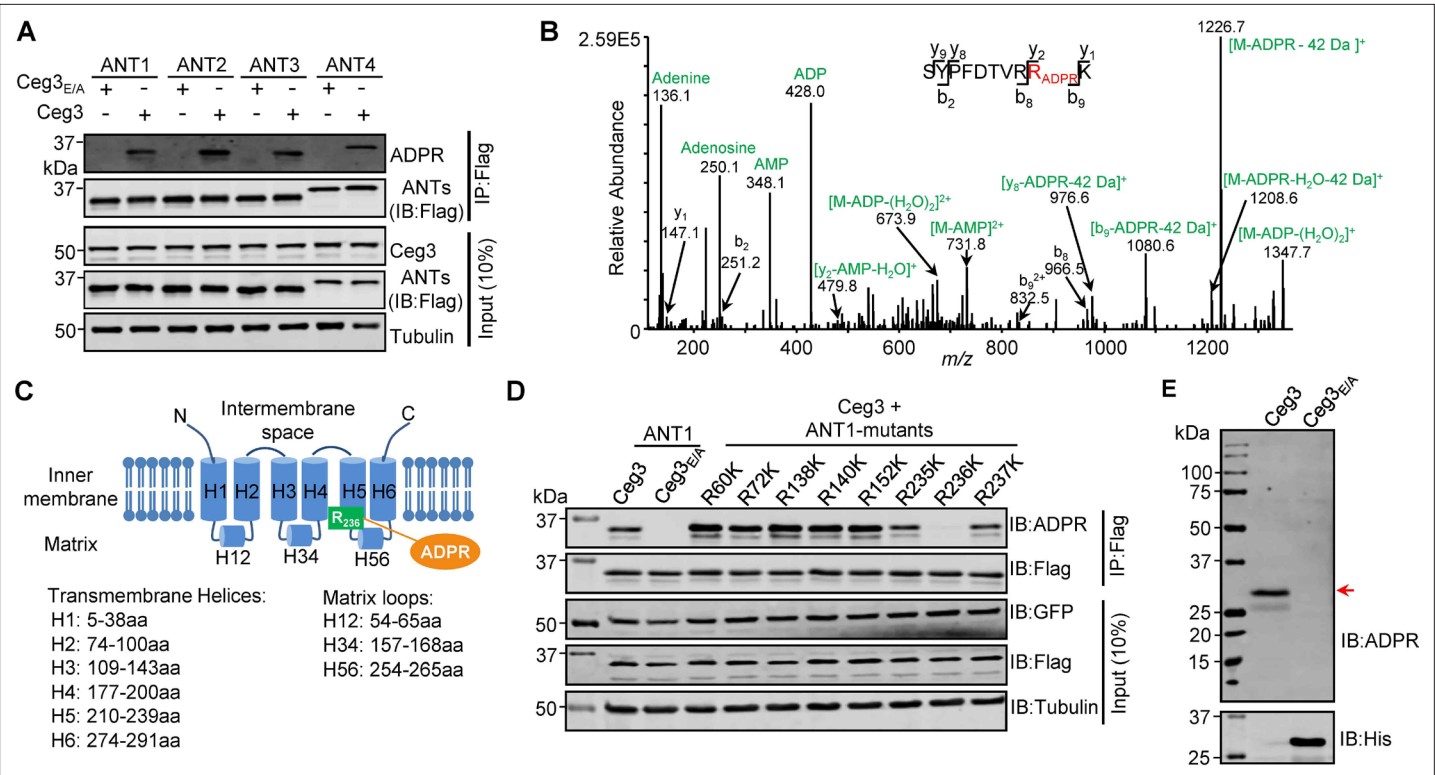

**Figure 3.** Determination of the modification sites on ADP/ATP translocases induced by Ceg3. (**A**) Ceg3 attacks all four ANT isoforms by ADP-ribosylation. Flag-tagged ANTs isolated from lysates of cells co-expressing GFP-Ceg3 or GFP-Ceg3$_{E/A}$ by immunoprecipitation were probed with antibodies specific for ADPR and Flag (top). The expression of relevant proteins was probed in total cell lysates with antibodies specific for GFP (Ceg3), Flag (ANTs), and Tubulin (bottom), respectively. (**B**) Mass spectrometric analysis of ADP-ribosylated 4xFlag-ANT1$_{V227K/R237K}$. Mono-ADP-ribosylation modification was detected in the peptide -S$_{228}$YPFDTVRRK$_{237}$-. Tandem mass (MS/MS) spectrum shows the fragmentation pattern of the modified peptide, with many ADP-ribosylation-specific marker ions and neutral loss fragments highlighted in green. (**C**) The schematic topology of ANT1 based on the structure of the bovine ADP/ATP carrier (PDB: 1OKC). Note the positioning of Arg236 at the end of helix 5. (**D**) Mutation of R236 but not neighboring arginine residues in ANT1 abolished Ceg3-induced modification. In each case, Flag-ANT1 or its mutants isolated from HEK293T cells co-expressing GFP-Ceg3 or GFP-Ceg3E/A were probed for ADPR modification (top) or for protein levels in IP products. The expression of Ceg3 and ANT1 or their mutants was probed in total lysates (input). Tubulin was detected as a loading control. (**E**) Ceg3 induces ADP-ribosylation modification of yeast ADP/ATP carriers. Lysates of yeast cells expressing His$_6$-Ceg3 or His$_6$-Ceg3$_{E/A}$ driven by a galactose-inducible promotor were detected for ADPR modification (top) (red arrow) or for the expression of Ceg3 (bottom). Note that the expression of wild-type Ceg3 was not detectable despite the presence of strong modification signals.

The online version of this article includes the following figure supplement(s) for figure 3:

**Figure supplement 1.** Lys substitutions of V227 and R237 in ANT1 do not affect Ceg3-induced ADP-ribosylation.

**Figure supplement 2.** Sequence alignment of the four human ADP/ATP isoforms.

**Figure supplement 3.** Recombinant Ceg3 expressed in *Escherichia coli* is insoluble.

**Figure supplement 4.** ANT1 is ADP-ribosylated by Ceg3 in *Escherichia coli*.

Human cells express four ADP/ATP translocase isoforms, each was identified in our mass spectrometric analysis (*Supplementary file 1*). ANTs are essential for transferring ADP and ATP across the mitochondrial inner membranes (*Klingenberg, 2008*), which is consistent with our finding that Ceg3 is targeted to this organelle (*Figure 1C–F*). Collectively, these results indicate that ANTs are the potential targets of Ceg3.

## Ceg3 ADP-ribosylates ANTs on an arginine residue in a conserved element

To examine whether Ceg3 targets all ANT isoforms by ADP-ribosylation, we co-expressed Flag-tagged each of the four isoforms (ANT1, ANT2, ANT3, and ANT4) with GFP-Ceg3 or GFP-Ceg3$_{E/A}$ in HEK293T cells and isolated Flag-ANTs proteins from cell lysates by IP. Detection using the ADP-ribose

probe revealed that each of the ANTs was modified by Ceg3 but not the mutant Ceg3$_{E/A}$ (*Figure 3A*). These results demonstrate that Ceg3 modifies these ADP/ATP translocases by ADP-ribosylation.

We next used ANT1 as the model to determine the site of modification. Mass spectrometric analysis of samples using wild-type ANT1 suggested that the modification likely occurs around Arg237 but the exact site could not be precisely assigned due to the large size of the peptides produced by protease trypsin or Lys-C from this region. We thus replaced Val227 and Arg237 with Lys in order to generate peptides of sizes more suitable for detection upon Lys-C digestion. The introduction of these mutations did not affect Ceg3-induced modification of ANT1 (*Figure 3—figure supplement 1A*). When ANT1$_{V227K/R237K}$ purified from cells co-expressing Ceg3 or Ceg3$_{E/A}$ (*Figure 3—figure supplement 1B*) were analyzed by mass spectrometry analysis, a mass shift of 541.06 Da matching to mono-ADP-ribosylation on the peptide -S$_{228}$YPFDTVRRK$_{237}$- in ANT1$_{V227K/R237K}$ was detected only in samples co-expressing Ceg3. MS/MS spectra showed that the modification mapped to residue Arg236, and the modified peptide produced several fragmented ions with *m/z* of 136.1, 250.1, 348.1, and 428.0 which are adenine, adenosine, adenosine monophosphate (AMP), and adenosine diphosphate (ADP), respectively, all are diagnostic fragments of the ADPR moiety (*Figure 3B*; *Rosenthal et al., 2015*). We also detected a series of fragments containing the conversion of arginine to ornithine residues (–42 Da), which is also diagnostic tandem mass fragment of ADP-ribosylated arginine residues (*Gehrig et al., 2021*). Thus, Ceg3 modifies ANT1 by mono-ADP-ribosylation on residue Arg236.

The structure of the bovine ADP/ATP carrier (>90% identity to those from humans) (*Pebay-Peyroula et al., 2003*) reveals that ANTs are consisting of six transmembrane helices and three matrix loops (*Figure 3C*). In ANT1, the modified site Arg236 resides in the end of the fifth transmembrane region, which is close to the matrix side of the mitochondrion (*Pebay-Peyroula et al., 2003*). Sequence alignment of the four mammalian ANT isoforms shows that the site modified by Ceg3 mapped to the second Arg residue of the conserved -RRRMMM- element (*Figure 3—figure supplement 2*), which has been shown to be important for the transport activity of ANTs (*Pebay-Peyroula et al., 2003*). We next confirmed the modification site in ANT1 by testing mutants with lysine substitutions in each of several conserved Arg residues (i.e., Arg60, Arg72, Arg138, Arg140, and Arg152) that are close to the interface between the transmembrane helices and the matrix loops of ANT1 and each of the three Arg residues (i.e., Arg235, Arg236, and Arg237) in the -R$_{235}$RRMMM$_{240}$- motif. Only mutations in Arg236 abolished Ceg3-induced ADP-ribosylation (*Figure 3D*). These results establish that Ceg3 specifically modifies Arg236 in the conserved -R$_{235}$RRMMM$_{240}$ motif in ANT1.

ADP/ATP carriers are evolutionarily conserved in eukaryotes, which in *S. cerevisiae* are represented by three isoforms, Aac1p, Aac2p, and Aac3p. These proteins have molecular weights similar to those of their mammalian counterparts and each harbors the -RRRMMM- element (*Ruprecht et al., 2014*). Given that Ceg3 inhibits the growth of yeast in an mART motif-dependent manner, we determined whether Ceg3 induces ADP-ribosylation of yeast ADP/ATP carrier proteins. Although the expression of His$_6$-Ceg3 in yeast cells was not detectable with a His$_6$-specific antibody, ADP-ribosylation of proteins with a molecular weight close to those of ADP/ATP carriers was detected in lysates of yeast cells expressing Ceg3 but not Ceg3$_{E/A}$ (*Figure 3E*). Thus, yeast ADP/ATP carriers are also targeted by Ceg3 for ADP-ribosylation modification, which likely accounts for its toxicity.

We attempted to establish biochemical reactions to study Ceg3-induced ADP-ribosylation of ANTs by preparing recombinant Ceg3 from *Escherichia coli*. Yet, none of the commonly used tags allowed the production of soluble Ceg3 under various induction conditions (*Figure 3—figure supplement 3A*) as well as several truncated mutants of Ceg3 (*Figure 3—figure supplement 3B*). The insoluble property of Ceg3 is consistent with the fact that Ceg3 is not a peripheral protein of the mitochondria (*Figure 1E*).

We then purified Flag-GFP-Ceg3 and Flag-GFP-Ceg3$_{E/A}$ from transfected mammalian cells and tested their ability to modify similarly purified Flag-ANTs by ADP-ribosylation. Although endogenous ANTs in cells expressing Flag-GFP-Ceg3 were modified, incubation of purified Ceg3 and ANT1 with NAD did not lead to detectable modification (*Figure 3—figure supplement 4A*). Importantly, when co-expressed in *E. coli*, Ceg3-induced ANT1 modification was detectable, again in a manner that requires the mART motif (*Figure 3—figure supplement 4B*). The ability of Ceg3 to modify ANT1 in *E. coli* indicates that this enzyme functions without the need of co-factors from eukaryotic cells.

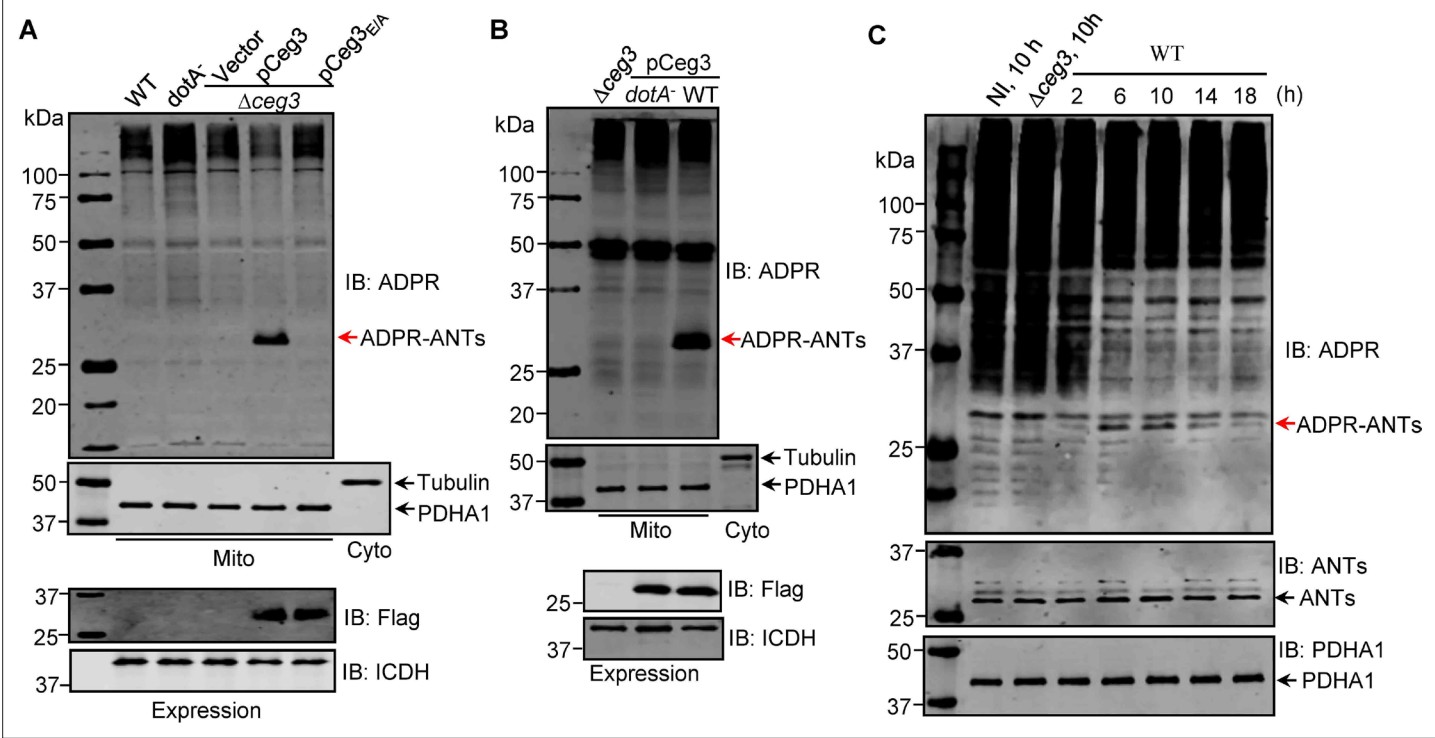

**Figure 4.** ADP-ribosylation of ADP/ATP translocases by Ceg3 occurs in cells infected with *Legionella pneumophila*. (**A**) An intact mART motif in Ceg3 is required for its modification of ADP/ATP translocases during *L. pneumophila* infection. Bacteria of the indicated *L. pneumophila* strains were opsonized prior to infecting HEK293T cells transfected to express the FcγII receptor at an MOI of 100 for 2 hr. Samples of the mitochondrial fraction were probed for ADPR after SDS-PAGE (top). PDHA1 and tubulin were probed as controls to monitor the success of cell fractionation. One cytosolic fraction sample was included as an additional control (middle). The expression of Flag-Ceg3 in bacteria was analyzed with the Flag antibody, the metabolic enzyme isocitrate dehydrogenase (ICDH) was probed as a loading control (bottom). (**B**) A functional Dot/Icm system is required for Ceg3-induced ADP-ribosylation of ADP/ATP translocases in infected cells. HEK293T cells expressing the FcγII receptor were infected with opsonized bacteria expressing Flag-Ceg3 at an MOI of 100 for 2 hr. Isolated mitochondrial proteins resolved by SDS-PAGE were probed for ADPR modification (top). The quality of cell fractionation was determined by probing for PDHA1 and tubulin (middle), respectively. The expression of Ceg3 in bacteria was detected with the Flag antibody, and ICDH was probed as a loading control (bottom). (**C**) ADP-ribosylation of ADP/ATP translocases is detectably induced by wild-type *L. pneumophila* at 6 hr post-infection. HEK293T cells expressing the FcγII receptor were infected with opsonized bacteria for the indicated periods of time at an MOI of 30. Mitochondrial proteins were analyzed by anti-ADPR and anti-PDHA1 Western blot. MOI, multiplicity of infection; NI, no infection.

The online version of this article includes the following figure supplement(s) for figure 4:

**Figure supplement 1.** Deletion of *ceg3* did not detectably affect intracellular growth of *Legionella pneumophila*.

## The ADP/ATP carriers are ADP-ribosylated by Ceg3 in cells infected with *L. pneumophila*

To examine whether the observed modification of ANTs by Ceg3 occurs under physiological conditions, we attempted to detect ADP-ribosylation of ANTs during *L. pneumophila* infection. It was clearly detectable in samples infected with a strain that lacks the chromosomal *ceg3* but expresses the gene from a multicopy plasmid (*Figure 4A*). ANTs modification was also detected in cells infected with the wild-type strain harboring the plasmid but not in a mutant defective in the Dot/Icm transporter (*Figure 4B*). Modification of ANTs was not detected by wild-type strain at 2 hr post-infection (psi) (*Figure 4A*), which may be due to the temporal regulation. We then extended the infection time to 18 hr and sampled infected cells at 4 hr intervals. Under this experimental condition, ADP-ribosylation of ANTs induced by wild-type *L. pneumophila* became detectable at 6 hr after bacterial uptake and peaked at 10 hr. Furthermore, the level of modification decreased at 14 hr and was maintained at low level till 18 hr (*Figure 4C*), which corresponds to maximal intracellular replication in this host cell (*Derré and Isberg, 2004*). This modification pattern suggests that *L. pneumophila* temporally regulates ANTs by ADP-ribosylation either by differently controlling the expression of Ceg3 at different phases of infection or by injecting enzymes capable of reversing the modification at infection phases

beyond 10 hr. Taken together, these results indicate that Ceg3 induces ADP-ribosylation of ANTs during *L. pneumophila* infection. The inability to detect modification at 2 hr psi may be due to low abundancy of translocated Ceg3 or modified ANTs have not reached the quantity detectable by the method we used.

We also examined the role of Ceg3 in intracellular replication of *L. pneumophila* using the Δ*ceg3* mutant in mouse bone marrow-derived macrophages (BMDMs) and the protozoan host *Dictyostelium discoideum*. Akin to most Dot/Icm substrates that are dispensable for intracellular replication in laboratory infection models, the Δ*ceg3* mutant grew at rates indistinguishable to those of the wild-type strain (*Figure 4—figure supplement 1*), indicating that the absence of Ceg3 does not detectably affect the intracellular replication of *L. pneumophila* in commonly used laboratory hosts.

## Ceg3 inhibits ANTs-mediated ADP/ATP exchange in mitochondria

The main role of mitochondrion is ATP production and regulation of metabolism flux, and the production of ATP entirely depends on the integrity of mitochondrial membrane potential (MMP) (*Zorova et al., 2018*). We thus examined whether Ceg3-induced ADP-ribosylation of ANTs compromises MMP. HEK293T cells transfected to express Ceg3 or its mutants for 24 hr were loaded with the JC-10 dye, which, in healthy mitochondria, forms aggregates that emits red fluorescence signals but diffuses out of mitochondria with damaged MMP and exhibits green fluorescence signals (*Reers et al., 1995*). As expected, treatment with the ionophore carbonyl cyanide m-chlorophenyl hydrazine (CCCP) that damages the MMP by uncoupling the proton gradient, rendered the cells to emit green fluorescence signals (*Figure 5A*; *Georgakopoulos et al., 2017*). The expression of Ceg3 but not Ceg3$_{E/A}$ caused a slight decrease in MMP. Thus, the impact of Ceg3 on mitochondrial membrane integrity, if any, is moderate.

ATP is transported across the inner mitochondrial membrane by ANTs via exchange with ADP at a 1:1 stoichiometry (*Pfaff et al., 1965*), we next examined the impact of Ceg3-induced modification on this by adding 2 mM ADP into mitochondria isolated from HEK293T cells transfected to express Ceg3 or its mutants and determined the amounts of released ATP (*Figure 5B*). Expression of Ceg3 but not Ceg3$_{E/A}$ in HEK293T cells caused a decrease in ADP/ATP exchange (*Figure 5C*). Moreover, such inhibition also occurred in mitochondria isolated from cells infected with *L. pneumophila* strains capable of inducing detectable ANT ADP-ribosylation (*Figure 5D*). Taken together, these results indicate that ADP-ribosylation of ANTs by Ceg3 blocks ADP/ATP exchange by mitochondria (*Figure 5E*).

## ANTs modified by ADP-ribosylation maintain their role in mitophagy induction

A recent study showed that ANT1 is critical for the induction of mitophagy (*Hoshino et al., 2019*), which is a branch of autophagy involved in the degradation of damaged mitochondria (*Youle and Narendra, 2011*). We thus examined whether Ceg3-induced ADP-ribosylation of ANTs affects their roles in mitophagy. After testing a few cell lines, we found that COS-1 cells undergo robust mitophagy upon being treated with the protonophores CCCP, which caused a clear loss of two mitochondrial proteins, PDHA1 and ATPB (*Figure 5—figure supplement 1A*). To ensure that Ceg3 is expressed in a high percentage of cells in these samples, we prepared lentiviral particles that direct Ceg3 expression. Transduction of COS-1 cells with our lentiviral particles led to mART-dependent ADP-ribosylation of endogenous ANTs (*Figure 5—figure supplement 1B*). Under our experimental conditions in which cells were uniformly transduced and the expression of Ceg3 is readily detectable, the levels of PDHA1 were similar among all samples (*Figure 5—figure supplement 1C*), suggesting that Ceg3 did not induce mitophagy. We also tested the effect of Ceg3 on CCCP-induced mitophagy. While treatment with this protonophore did lead to a drastic decrease in PDHA1, such decrease did not detectably change between samples expressing Ceg3 or Ceg3$_{E/A}$ (*Figure 5—figure supplement 1C*). Thus, Ceg3 does not suppress or augment CCCP-induced mitophagy.

## Discussion

In addition to powering most cellular activities by ATP production, mitochondria are involved in diverse functions such as cell death, immunity, and metabolism regulation (*McBride et al., 2006*). As a result, this organelle is a common target for infectious agents that actively manipulate host cellular

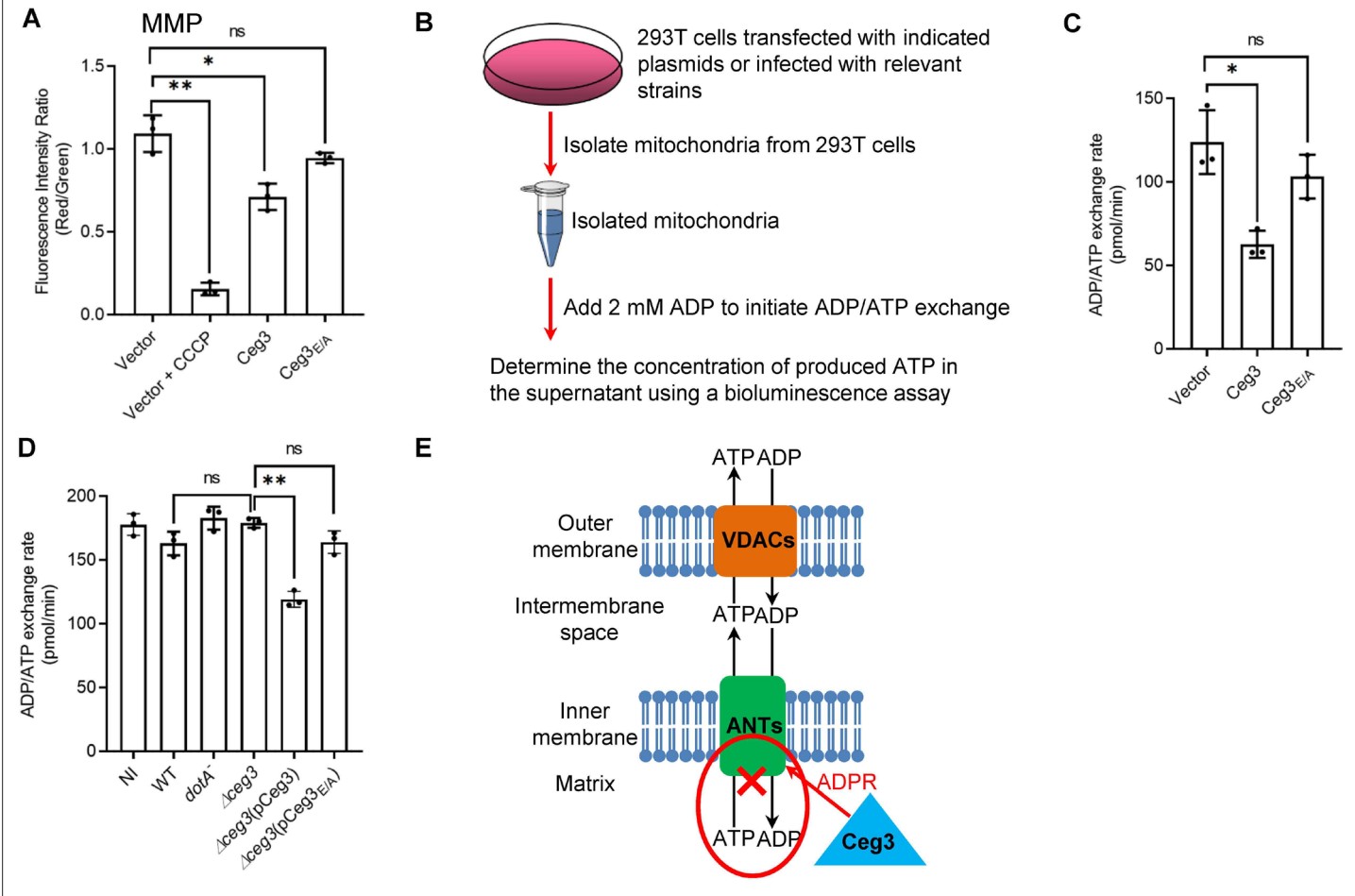

**Figure 5.** Ceg3 inhibits ADP/ATP exchange in mitochondria. (**A**) Ceg3 interferes with the mitochondrial membrane potential (MMP). HEK293T cells transfected to express Ceg3 or its inactive mutant Ceg3$_{E/A}$ were used to determine MMP by the JC-10 dye. Samples treated with 20 µM CCCP for 1 hr were included as a positive control for loss of MMP integrity. Quantitation shown was from three independent experiments done in triplicate. Error bars: standard error of the mean (SEM). Statistical analysis was determined by two-tailed t-test. ns, not significant; *, p<0.05; **, p<0.01. (**B**) The workflow for measuring ADP/ATP exchange rates. (**C**) Ceg3 interferes with the ADP/ATP exchange by mitochondria. Mitochondria isolated from HEK293T cells expressing the indicated proteins were suspended in a reaction buffer containing 10 mM HEPES (pH 7.4), 250 mM sucrose, and 10 mM KCl. 2 mM ADP was added to initiate the ADP/ATP exchange process. After 5 min incubation, the concentrations of ATP transported from mitochondria were determined to calculate the ADP/ATP exchange rates. Quantitation shown was from three independent experiments. Error bars: standard error of the mean (SEM). Statistical analysis was determined by two-tailed t-test. ns, not significant; *, p<0.05. (**D**) Ceg3 perturbs ADP/ATP exchange in cells infected with *Legionella pneumophila*. Opsonized bacteria of the indicated *L. pneumophila* strains were used to infect HEK293T cells expressing the FcγII receptor at an MOI of 100 for 2 hr. Mitochondria isolated from the infected cells were used to determine ADP/ATP exchange rates. Results shown were from three independent experiments. Error bars: standard error of the mean (SEM). Statistical analysis was determined by two-tailed t-test. ns, not significant; *, p<0.05; **, p<0.01. (**E**) A diagram depicting the inhibition of mitochondrial ADP/ATP exchange by Ceg3. Ceg3-induced modification of ANTs by ADPR in the inner membrane blocks the ADP/ATP transport activity of the translocases. MOI, multiplicity of infection; NI, no infection.

The online version of this article includes the following figure supplement(s) for figure 5:

**Figure supplement 1.** ADP-ribosylation of ANTs by Ceg3 does not affect mitophagy induction.

processes for their survival and replication (***Lobet et al., 2015***). For example, the *V. cholera* T3SS effector VopE inhibits the activity of Miro1 and Miro2, two mitochondrial GTPases, by functioning as a GTPase-activating protein (GAP), thus suppressing host innate immune responses (***Suzuki et al., 2014***). The effector ECH0825 from *Ehrlichia chaffeensis* upregulates the mitochondrial manganese superoxide dismutase to prevent ROS-induced cellular damage and mitochondria-mediated apoptosis (***Liu et al., 2012***).

Earlier cell biological studies suggest that mitochondrion plays important roles in the intracellular life cycle of *L. pneumophila*. The association of the LCV with mitochondria was documented in

morphological analysis of its infection cycle within several years after the bacterium was recognized as a pathogen. At least 30% of LCVs were surrounded by mitochondria within 15 min of phagocytosis, which increases to approximately 65% when the infection proceeds to 1 hr (**Horwitz, 1983**). This observation was validated by more recent studies, which reveal that such association requires the Dot/Icm transporter essential for *L. pneumophila* virulence (**Tilney et al., 2001**).

A number of Dot/Icm effectors have been demonstrated to modulate the function of mitochondria. Among these, LegS2, a homolog of the eukaryotic enzyme sphingosine-1-phosphate lyse, contributes to autophagy inhibition by disrupting lipid metabolism in mitochondria (**Degtyar et al., 2009**; **Rolando et al., 2016**). MitF appears to modulate mitochondrial dynamics by provoking a Warburg-like effect to benefit intracellular bacterial replication (**Escoll et al., 2017**). Our results here demonstrate that Ceg3 specifically localizes to mitochondria where it modifies members of the ADP/ATP translocase by mono-ADP-ribosylation, leading to inhibition of ADP/ATP exchange in mitochondria.

ADP-ribosylation of ANTs induced by Ceg3 is readily detectable in yeast and mammalian cells that are ectopically expressing the effector (**Figures 2A and 3E**). Due to technical barriers to obtain soluble recombinant Ceg3, we were unable to establish a reaction with purified proteins to show Ceg3-induced ADP-ribosylation of ANTs. The fact that such modification occurs when Ceg3 and ANT1 were co-expressed in *E. coli* indicates that no host cell co-factor is required for its activity (**Figure 3—figure supplement 4B**). The detection of ANT modification in cells infected with wild-type *L. pneumophila* further validates these translocases as its bona fide substrates.

The ADP-ribosylation site on ANT1 induced by Ceg3 is Arg236, which is the second Arg residue in the -RRRMMM- element conserved in not only all four ADP/ATP translocases of mammalian origin, but also all known ADP/ATP carriers from different organisms (**Körver-Keularts et al., 2015**). The three arginine residues in the -RRRMMM- motif are essential for the transport function of the yeast ADP/ATP carrier Aac2 (**Müller et al., 1996**; **Nelson et al., 1993**). Mutations in each of these three arginine residues in the bovine ADP/ATP carrier abolish its transport activity, so does a mutation in Arg246 (equivalent to Arg236 in mammalian ANT1) of the mitochondrial carrier from *Thermothelomyces thermophile* (**Pebay-Peyroula et al., 2003**; **Ruprecht et al., 2019**). These observations are consistent with our results that ADP-ribosylation of one of these three sites ablates its ADP/ATP transport activity (**Figure 5C–D**). Moreover, our finding that Ceg3 does not interfere with mitophagy is in line with a previous study showing that the ADP/ATP translocase-associated mitophagy is independent of its nucleotide translocase activity (**Hoshino et al., 2019**).

Structure of the bovine ADP/ATP carrier shows that Arg236 of ANT1 is localized in the end of the fifth transmembrane helix, which is in close proximity to the mitochondrial matrix (**Pebay-Peyroula et al., 2003**). In the Aac2 carrier from yeast, the -RRRMMM- element appears to be in the mitochondrial matrix (**Ruprecht et al., 2014**). Although the positioning of the -RRRMMM- element revealed by these two structural analyses slightly differs, both studies conclude that the second arginine is implicated in ADP binding via its positively charged side chain that mediates electrostatic interactions with the negatively charged phosphate moieties in ADP (**Pebay-Peyroula et al., 2003**; **Ruprecht et al., 2014**). At neutral pH, ADP-ribose harbors two negatively charged phosphate moieties, which can cause a notable change in the charge property of the side chain of this arginine residue (**Laing et al., 2011**). Thus, Ceg3-mediated ADP-ribosylation on ANTs likely affects the electrostatic interactions with ADP. In addition, the bulky ADPR moiety on ANTs may cause steric interference for the interaction, both of which can contribute to the inhibition of their ADP/ATP exchange ability.

The Dot/Icm effector LncP, which is homologous to mitochondrial carrier proteins, has been suggested to carry out unidirectional ATP transport across membranes reconstituted in liposomes (**Dolezal et al., 2012**). Although the possibility that Ceg3 coordinates its activity with LncP in *L. pneumophila* infected cells needs further investigation, it is conceivable that the bacterium can gain better control of ATP transport in the mitochondria if it uses one set of effectors to inhibit the endogenous carriers and another set to substitute this function. The number of Dot/Icm effectors that function to modulate mitochondrial activity seems large. An earlier study found that Lpg1625 and Lpg0898 are targeting to this organelle (**Zhu et al., 2013**). Lpg2444 was recently shown to protect the integrity of mitochondria by making it resistant to MMP damage caused by protonophores such as CCCP. This protein also interacts with ADP/ATP translocase 2, but the biological significance of such interactions remains elusive (**Noack et al., 2020**). Intriguingly, although these effectors clearly are targeted to the mitochondria, none of them harbors a discernable canonical mitochondrial localization signal (MLS)

or mitochondrial targeting sequence (MTS) found in eukaryotic proteins (*Habib et al., 2007*). Future study aiming at biochemical analysis of these proteins, their cellular substrates, and mechanism of organelle targeting will shed light on the roles of mitochondria in *L. pneumophila* virulence as well as the mechanism of how the diverse activities of this organelle are regulated.

## Materials and methods
### Media, bacteria strains, plasmid construction, and cell lines
*E. coli* strains were grown on LB agar plates or in LB broth. When necessary, antibiotics were added to media at the following concentrations: ampicillin, 100 µg/ml; kanamycin, 30 µg/ml. *L. pneumophila* strains used in this study were derivatives of the Philadelphia 1 strain Lp02 (*Berger and Isberg, 1993*). Lp03 is an isogenic *dotA⁻* mutant (*Berger et al., 1994*). All strains were grown and maintained on CYE plates or in ACES-buffered yeast extract (AYE) broth as previously described (*Berger and Isberg, 1993*). For *L. pneumophila,* antibiotics were used at: kanamycin, 20 µg/ml, streptomycin, 100 µg/ml. When needed, thymidine was added at a final concentration of 100 µg/ml. Other methods are available in Supporting Information.

### Plasmid construction and cell lines
The oligonucleotides, plasmids, and bacterial strains used in this study were listed in *Supplementary file 2*. The Δceg3 in-frame deletion strain was constructed by a two-step allelic exchange strategy as described (*Duménil and Isberg, 2001*). For complementation experiments, genes were inserted into pZLQ-Flag, a derivative of pZLQ (*Luo and Farrand, 1999*) that was modified to carry a Flag tag. For ectopic expression of proteins in mammalian cells, genes were inserted into pEGFPC1 (Clontech), the 4xFlag CMV vector (*Li et al., 2021*) or pAPH, a derivative of pVR1012 (*Wang et al., 2018*) suitable for expressing proteins with an amino HA tag and a carboxyl Flag tag. To co-express Ceg3 and ANT1 in *E. coli*, *ceg3*, and *ANT1* were inserted into pZLQ-Flag and pGEX-6p-1 (GE Healthcare), respectively. The integrity of all constructs was verified by sequencing analysis. HEK293T, HeLa, or COS-1 cells purchased from ATCC were cultured in Dulbecco's modified minimal Eagle's medium (DMEM) supplemented with 10% fetal bovine serum (FBS). BMDMs were prepared from 6- to 10-week-old female A/J mice (Jackson Lab) with L-cell supernatant-conditioned medium as described previously (*Conover et al., 2003*). All mammalian cell lines were regularly checked for potential mycoplasma contamination by the universal mycoplasma detection kit from ATCC (Cat# 30-1012K).

### Yeast toxicity assays
All yeast strains used in this study were derived from W303 *Tan et al., 2011*; yeast was grown at 30°C in YPD medium or in appropriate amino acid dropout synthetic media supplemented with 2% of glucose or galactose as the sole carbon source. Yeast transformation was performed as previously described (*Gietz et al., 1995*). Inducible protein toxicity of Ceg3 was assessed by the galactose-inducible promoter on pYES1NTA (Invitrogen). Briefly, plasmids harboring Ceg3 or its mutant derived from pYES1NTA were transformed into yeast strain W303. Yeast grown in liquid selective medium in the presence of glucose was serially diluted fivefold and 5 µl of each dilution was spotted onto selective plates containing glucose or galactose. Plates were incubated at 30°C for 3 days before image acquisition.

### Carbonate treatment of mitochondria
Mitochondria were isolated from cultured cells using a mitochondria isolation kit (Thermo Fisher Scientific, cat# 89874) according to the manufacturer's instructions. Isolated mitochondria were resuspended in a high pH carbonate buffer (0.1 M $Na_2CO_3$, pH 11) on ice for 30 min with occasional agitation. Samples were centrifuged at 15,000×*g* for 20 min at 4°C. Integral membrane proteins were collected in pellet fraction, while peripheral membrane proteins and soluble proteins were harvested in supernatant fraction. Both fractions were resolved by SDS-PAGE and analyzed by immunoblotting.

### Transfection, immunoprecipitation, infection
Plasmids were transfected into mammalian cells by using Lipofectamine 3000 (Invitrogen, cat# L3000150). After 24 hr transfection, cells were collected and lysed with the TBS buffer (150 mM NaCl,

50 mM Tris-HCl, pH 7.5) with 1% Triton X-100. When needed, IP was performed with lysates of transfected cells by using beads coated with the Flag antibody (Sigma-Aldrich, cat# F2426) at 4°C overnight. Beads were washed with pre-cold lysis buffer for three times. All samples were resolved by SDS-PAGE and followed by immunoblotting analysis with the specific antibodies. For ADP-ribosylated proteins enrichment by Af1521 pulldown, Affigel beads (Bio-Rad) were coated with recombinant His-Af1521 proteins with protocols supplied by the manufacturer and then were incubated with lysates of HEK293T cells transfected to express Ceg3 or Ceg3$_{E/A}$ overnight at 4°C. Affigel beads were washed three times and bound proteins were treated with SDS sample buffer. Proteins resolved by SDS-PAGE were visualized by silver staining and immunoblotting.

For all *L. pneumophila* infection experiments, *L. pneumophila* strains were grown in AYE broth to the post-exponential phase judged by optical density (OD$_{600nm}$=3.2–4.0) as well as increase in motility. Complementation strains were induced with 0.2 mM isopropyl β-D-1-thiogalactopyranoside (IPTG) for 4 hr at 37°C before infection. To determine the modification of ANTs and ADP/ATP exchange rates in mitochondria during bacterial infection, HEK293T cells transfected to express FCγRII receptor (*Qiu et al., 2016*) for 24 hr were infected with opsonized bacteria at a multiplicity of infection (MOI) of 100. Two hours after infection, mitochondria were isolated from cells for further experiments. For assays of *L. pneumophila* growth within, approximately $4\times10^5$ BMDMs or *D. discoideum* seeded into 24-well plates 1 day before infection were infected with relevant *L. pneumophila* at an MOI of 0.05. Two hours after adding the bacteria, we synchronized the infection by washing the monolayers three times with phosphate-buffered saline (PBS) buffer. Infected macrophages or *D. discoideum* were incubated at 37°C in the presence of 5% CO$_2$ or at 25°C, respectively. At each time point, cells were lysed with 0.02% saponin, dilutions of the lysate were plated onto bacteriological media, and CFU was determined from triplicate wells of each strain.

## Protein purification

For His-Af1521 protein production, 10 ml overnight *E. coli* cultures were transferred to 400 ml LB medium in the presence of 100 μg/ml ampicillin and grown to OD$_{600nm}$ of 0.6–0.8. Then the cultures were incubated at 18°C for 16–18 hr after the addition of IPTG at a final concentration of 0.2 mM. Bacterial cells were harvested at 12,000×$g$ by spinning and lysed by sonication. The soluble lysates were cleared by spinning at 12,000×$g$ twice at 4°C for 20 min. His-tagged proteins were purified with Ni$^{2+}$-NTA beads (QIAGEN) and were eluted with 300 mM imidazole in PBS buffer. Purified proteins were dialyzed in buffer containing PBS, 5% glycerol, and 1 mM DTT overnight.

## Antibodies and immunoblotting

For immunoblotting, samples resolved by SDS-PAGE were transferred onto 0.2 μm nitrocellulose membranes (Bio-Rad, cat# 1620112). Membranes were blocked with 5% non-fat milk, and then incubated with the appropriate primary antibodies: anti-HA (Sigma-Aldrich, cat# H3663), 1:5000; anti-Flag (Sigma-Aldrich, cat# F1804), 1:5000; anti-ICDH (*Xu et al., 2010*), 1:10,000; anti-tubulin (DSHB, E7) 1:10,000; anti-PDHA1 (ProteinTech, cat# 18068-1-AP), 1:5000; anti-ATPB (ProteinTech, cat# 17247-1-AP), 1:2000; anti-TOM20 (ProteinTech, cat# 11802-1-AP), 1:5000; anti-VDAC1 (ProteinTech, cat# 55259-1-AP), 1:2000; anti-Cyto c (Santa Cruz Biotechnology, cat# sc-13560), 1:1000; anti-Calnexin (ProteinTech, cat# 10427-2-AP), 1:2000; anti-ANT1/2 (ProteinTech, cat# 17796-1-AP), 1:1000; anti-GM130 (BD Biosciences, cat# 610822), 1:2500; and anti-ADPR (Sigma-Aldrich, cat# MABE1016), 1:1000. Membranes were then incubated with an appropriate IRDye infrared secondary antibody and scanned by using an Odyssey infrared imaging system (Li-Cor's Biosciences).

## Immunostaining

Hela Cells were seeded at $5\times10^4$ per well on glass coverslips in 24-well plates 1 day before transfection. Cells were transfected to express corresponding proteins for 18 hr, and then the medium was replenished with fresh medium containing 500 nM Mitotracker Red (Thermo Fisher Scientific, cat# M22425). After incubated at 37°C for 30 min, cells were fixed by 4% formaldehyde solution for another 30 min at 4°C. Fixed cells were permeabilized by 0.2% Triton X-100 solution for 5 min, and blocked with 4% goat serum for 30 min at 37°C. Flag-tagged proteins were stained with a specific antibody (Sigma-Aldrich, cat# F1804) at a dilution of 1:50. Incubation with primary antibodies was performed overnight at 4°C, and then cells were stained with secondary antibodies conjugated to

Alexa Flour 594 or Alexa Flour 488 (Thermo Fisher Scientific) at a dilution of 1:500 for 1 hr at room temperature. After staining for nucleus with Hoechst, images were acquired using a Zeiss LSM 880 confocal microscope. The colocalization of Ceg3 with mitochondria was quantitated by measuring the Pearson correlation coefficient values with the ImageJ software (http://rsb.info.nih.gov/ij/).

## Lentiviral transduction

To produce lentivirus for transduction expression of Ceg3 in COS-1 cells, Ceg3 was inserted into pCDH-CMV-MCS-EF1a-RFP (System Biosciences cat# CD512B-1), which was transfected together with pMD2.G (Addgene plasmid #12259) and psPAX2 (Addgene plasmid #12260) vector (*Salmon and Trono, 2007*) into HEK293T cells grown to about 70% confluence. Supernatant was collected after 48 hr and then filtered with a 0.45-µm syringe filter. The titer of the produced lentivirus was determined by using qPCR Lentivirus Titer Kit (abm, cat# LV900). For lentiviral transduction, approximately $1 \times 10^5$ COS-1 cells seeded into 24-well plates 1 day before transduction were transduced with lentiviral particles at an MOI of 10. Cells incubated for 2 days at 37°C with 5% $CO_2$ were collected for immunoblotting.

## Detection of mitochondrial membrane potential

The MMP was measured using a mitochondria membrane potential kit from Sigma (Cat# MAK159) as previously described (*Wang et al., 2019*). Briefly, approximately $5 \times 10^4$ HEK293T cells were seeded in opaque 96-well plates with clear bottom 1 day before transfection. After 18 hr of transient expression, 50 µl JC-10 Dye loading solution was added to each sample well and incubated at 5% $CO_2$, 37°C for 1 hr, then 50 µl of assay buffer B was added. The intensity of red fluorescence ($\lambda_{ex}$=540/$\lambda_{em}$=590 nm) and green fluorescence ($\lambda_{ex}$=490/$\lambda_{em}$=525 nm) was monitored by a BioTek reader (Synergy 2, BioTek). The ratio of red/green fluorescence intensity was used to determine MMP. Samples treated by 20 µM CCCP for 1 hr was applied as a positive control of MMP loss.

## ADP/ATP exchange rates determination

Mitochondria isolated from one 10 cm plate of HEK293T cells transfected to express the proteins of interest or infected with relevant *L. pneumophila* strains were washed three times and resuspended in a reaction buffer (10 mM HEPES [pH 7.4], 250 mM sucrose, and 10 mM KCl). The ADP/ATP exchange process was initiated by the addition of ADP at a final concentration of 2 mM. After 5 min incubation, the amount of ATP transported from mitochondria was determined using an ATP measurement kit (Invitrogen, cat# A22066). Luminescence of samples was detected by a BioTek reader (Synergy 2, BioTek). For each experiment, a standard curve was generated with serially diluted ATP and was used to calculate the concentration of ATP in samples to determine ADP/ATP exchange rates.

## Immunogold labeling

HEK293T cells transiently expressing Flag-Ceg3 were washed with PBS briefly and fixed overnight with 4% formaldehyde and 0.5% glutaraldehyde in 0.1 M phosphate buffer (PB), pH 7.4 at 4°C. Cells were then washed with 0.1 M PB three times and then incubated in 0.1 M glycine solution (pH 2.2) for 20 min to quench the free aldehyde groups. The cell pellet was rinsed in 0.1 M PB buffer, and then dehydrated in an ascending ethanol series and infiltrated with LR white resin. Samples were embedded in gelatin capsules and polymerized. Polymerized samples were sectioned on an ultramicrotome and 60–70 nm thick sections were collected onto 200 mesh nickel grids. Grids were blocked with PBS containing 1% bovine serum albumin (BSA) (PBS-BSA buffer) for 30 min at 37°C and then incubated with the anti-FLAG M2 monoclonal antibody (Sigma, cat# F1804) at a 1:50 dilution in the PBS-BSA buffer overnight at 4 °C. The grids were rinsed with the same buffer five times and then were incubated with anti-mouse IgG conjugated with 10 nm gold particles (Sigma-Aldrich, cat# G7652) at a 1:10 dilution in 1% BSA-PBS for 1 hr. Grids were then washed with PBS and distilled water to remove unbound gold conjugate. The labeled samples were then post-stained with uranyl acetate and washed with distilled water prior to being examined by a Tecnai T12 (WSLR S046) transmission electron microscopy.

## LC-MS/MS analysis

Protein bands were digested in-gel with trypsin/Lys-C for protein identification or Lys-C for ADP-ribosylation modification site identification as previously described (*Shevchenko et al., 2006*). Digested peptides were injected onto a Waters NanoAcquity liquid chromatography (LC) system, coupled to Thermo Orbitrap Eclipse or Lumos. The LC has a dual pump configuration. Samples were desalted on a 4-cm reversed-phase trapping column (in-house packed, 150 µm i.d. 5 µm Jupiter C18 particle from Phenomenex) at 5 µl/min for 10 min. The analytical separation was on a 70-cm reversed-phase column (in-house packed, 75 µm i.d. 3 µm Jupiter C18 particle from Phenomenex) at 0.3 µl/min over 2 hr. Mobile phases are 0.1% formic acid in water for A, and 0.1% formic acid in acetonitrile for B. The gradient started at 1% B, and ramped to 8%–12%−30%–45%–95% B at 2-20-75-97-100 min, respectively.

MS source was set to 2.2 kV for electrospray, 250°C for capillary inlet, and RF lens at 30%. Acquisition method is data-dependent peptide mode with cycle time of 3 s. Isolation window was 1.6 *m/z*. Alternating higher-energy collisional dissociation (HCD) and electron transfer dissociation (ETD) were applied to the same precursor. HCD had stepped energy of 20%, 30%, and 40%. ETD reaction time follows the calibrated parameters, with supplemental HCD of 20%. Resolution setting was 60k for MS1 and 30k for MS2. Normalized AGC target was 250% for MS1, 100% for HCD, and 200% for ETD. Injection time control was set to auto.

For protein identification, the raw data were processed with the software Mascot (version 2.3.02, Matrix Science) against *Homo sapiens* database (Uniprot, UP000005640). Mascot was set to search with the following parameters: peptide tolerance at 0.05 Da, MS/MS tolerance at 0.2 Da, carbamidomethyl (C) as a fixed modification, oxidation (M) as a variable modification, and maximum of two missed cleavage. The false-discovery rates (FDRs) were controlled at <1%. To identify the ADP-ribosylation modification sites, data were analyzed using Byonic v3.11 (Protein Metrics). Protein FASTA contained the target protein and common contaminations (trypsin, keratin, etc., as provided in Byonic). Semi-specific and two max missed cleavages were allowed. Mass error tolerance was 7 ppm for precursor and 10 ppm for fragments. Dynamic modifications (provided in Byonic) include carbamidomethyl (+57.02 Da), oxidation (+15.99 Da), phosphorylation (+79.97 Da), and ADP-ribosylation (+541.06 Da). Custom modifications included phosphoribosylation (+212.01 Da), ADP (+409.02 Da), AMP (+329.20 Da), and ribosylation (+132.04 Da). Identified peptides were then examined manually for spectral quality. High confidence peptides with target modifications were further examined manually in Xcalibur QualBrowser.

## Data quantitation and statistical analyses

Student's t-test was used to compare the mean levels between two groups each with at least three independent samples. All western blot results shown are one representative from three independent experiments.

## Acknowledgements

The authors thank Dr. Shaohua Wang for plasmids, Dr. Victor Roman for making pZLQ-Flag, and Mr. Karl Weitz and Mr. Ronald Moore for assistance in mass spectrometry analysis. The authors thank Dr. Christopher J Gilpin, Dr. Robert Seiler, and Dr. Laurie Mueller for their support in TEM analysis which was performed at the Purdue Electron Microscopy Facility. This work in part was supported by the National Institutes of Health Grant R01AI127465 (ZQL) and by Jilin Science and Technology Agency Grant 20200403117SF (LS). Mass spectrometry analysis was performed in the Environmental Molecular Sciences Laboratory, a U.S. Department of Energy (DOE) national scientific user facility at Pacific Northwest National Laboratory (PNNL) in Richland, WA. Battelle operates PNNL for the DOE under Contract DE-AC05-76RLO01830.

## Additional information

### Funding

| Funder | Grant reference number | Author |
|---|---|---|
| National Institutes of Health | R01AI127465 | Zhao-Qing Luo |
| Jilin Science and Technology Agency | 20200403117SF | Lei Song |

The funders had no role in study design, data collection and interpretation, or the decision to submit the work for publication.

### Author contributions

Jiaqi Fu, Conceptualization, Formal analysis, Investigation, Methodology, Validation, Writing – original draft, Writing – review and editing; Mowei Zhou, Investigation, Methodology, Validation, Writing – review and editing; Marina A Gritsenko, Investigation, Methodology, Writing – review and editing; Ernesto S Nakayasu, Methodology, Resources, Writing – review and editing; Lei Song, Conceptualization, Formal analysis, Funding acquisition, Project administration, Supervision, Writing – review and editing; Zhao-Qing Luo, Conceptualization, Formal analysis, Funding acqui-sition, Project administration, Resources, Supervision, Validation, Writing – original draft, Writing – review and editing

### Author ORCIDs

Jiaqi Fu http://orcid.org/0000-0003-0081-6133
Lei Song http://orcid.org/0000-0002-4115-065X
Zhao-Qing Luo http://orcid.org/0000-0001-8890-6621

### Decision letter and Author response

Decision letter https://doi.org/10.7554/eLife.73611.sa1
Author response https://doi.org/10.7554/eLife.73611.sa2

## Additional files

### Supplementary files

• Supplementary file 1. Identification of ADP/ATP translocases as the targets of Ceg3 in samples obtained by Af1521-pulldown. The protein band specifically present in Af1521 pulldown samples from expressing Ceg3 was analyzed by mass spectrometry; the 10 proteins with the most hits were listed.

• Supplementary file 2. Bacterial strains, antibodies, plasmids and primers used in this study.

• Transparent reporting form

• Source data 1. Figures with the uncropped gels or blots with the relevant bands clearly labeled.

• Source data 2. The original files of the raw unedited gels or blots.

### Data availability

We have included all data to support our conclusions in the manuscript.

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
