## [Editor Report]

In this paper you show that the intracellular pathogen Legionella pneumophila uses an ADP-ribosyltransferase, Ceg3, to target mitochondria in infected cells. Ceg3 interferes with ADP/ATP exchange thereby modulating the energy metabolism in infected cells. This provides interesting new insight into the manipulation of mitochondrial function by an intracellular bacterium.

---

## [Decision Letter]

**Decision letter after peer review:**

Thank you for submitting your article "Legionella pneumophila modulates host energy metabolism by ADP-ribosylation of ADP/ATP translocases" for consideration by *eLife*. Your article has been reviewed by 2 peer reviewers, and the evaluation has been overseen by a Reviewing Editor and Dominique Soldati-Favre as the Senior Editor. The following individuals involved in review of your submission have agreed to reveal their identity: Hayley Newton (Reviewer #1); Elizabeth Hartland (Reviewer #2).

Essential revisions:

1. Improve the localization of Ceg3 by immunofluorescence microscopy. Co-localisation experiments with mitochondrial markers in Figure 1C and 2E should include imaging quantification e.g. Pearson's coefficient. It is also important to validate the mitochondrial localisation of Ceg3 that is secreted through the T4SS, can the authors detect FLAG-Ceg3 at mitochondria during infection (with L. pneumophila pCeg3 that was used in Figure 4)?

2. Figure 4C is missing a dceg3 control and immunoblotting for the total amount of ANT.

3. The timing of the detected modification of ANT should be discussed considering the divergence between Figure 4 and Figure 5: the infection data presented in 5D is done at 2h post-infection where ANT modification is not detected during WT infection (Figure 4A). Adding a sample of uninfected cells in 5D is needed to interpret the results better.

*Reviewer #1:*

The manuscript by Fu et al., demonstrates that the Legionella pneumophila Dot/Icm effector protein Ceg3 is a mono-ADP-ribosyltransferase that appears to be recruited to host cell mitochondria. Here Ceg3 is able to specifically ADP-ribosylate ADP/ATP translocases on the second arginine of the -RRRMMM- motif. This modification appears to render ADP/ATP translocases unable to exchange ADP/ATP across the inner mitochondrial membrane. The implications of this disruption to normal mitochondria function is not explored.

The biochemical characterisation of Ceg3 activity is a strength of this manuscript. In particularly the use of mass spectrometry to identify the site of ANT modification, demonstration that all four human ANT isoforms are modified by Ceg3 and the ability to link this Ceg3 function with previous observations about the impact Ceg3 has on Saccharamyces cerevisiae growth. The study also has enticing data regarding the temporal dynamics of ANT modification which may aid in clarifying both the role and impact this modification has during the context of infection. Further investigation into this aspect of this research will be of significant value to researchers in the host-pathogen interaction field.

*Reviewer #2:*

This manuscript identifies the Dot/Icm effector, Ceg3, from Legionella pneumophila as an ADP-ribosyltransferase that traffics to mitochondria in infected cells. The authors convincingly show that Ceg3 modifies the mitochondrial ADP/ATP carriers, which are located in the inner mitochondrial membrane, and interferes with ADP/ATP exchange.

The manuscript presents strong biochemistry supporting the authors' claims but improved microscopy is needed to support the mitochondrial location of Ceg3, and preferably its trafficking to the mitochondrial inner membrane. I think this work will be of interest to the broader research field studying mitochondrial biology and intracellular infection.

---

## [Author Response]

Essential revisions:1. Improve the localization of Ceg3 by immunofluorescence microscopy. Co-localisation experiments with mitochondrial markers in Figure 1C and 2E should include imaging quantification e.g. Pearson's coefficient. It is also important to validate the mitochondrial localisation of Ceg3 that is secreted through the T4SS, can the authors detect FLAG-Ceg3 at mitochondria during infection (with L. pneumophila pCeg3 that was used in Figure 4)?

We thank the reviewer for raising this important point. We have followed the suggestion to further prove mitochondrial localization of Ceg3 by using confocal microscope and performing imaging quantification by determining the Pearson correlation coefficient (Figures 1C and 2E). We have also determined the mitochondrial localization of Ceg3 during *L. pneumophila* infection by fractionation, which showed that targeting of Flag-Ceg3 to the mitochondria occurs only in cells infected with a strain with a functional Dot/Icm transporter (Figure 1F).

2. Figure 4C is missing a dceg3 control and immunoblotting for the total amount of ANT.

Thank you for pointing out this. We have redone the experiment by including a sample of cells infected with the ∆*ceg3* strain and also determined the total amount of ANTs by blotting with ANTs antibody in Figure 4C.

3. The timing of the detected modification of ANT should be discussed considering the divergence between Figure 4 and Figure 5: the infection data presented in 5D is done at 2h post-infection where ANT modification is not detected during WT infection (Figure 4A). Adding a sample of uninfected cells in 5D is needed to interpret the results better.

As suggested, we have included a sample of uninfected cells in Figure 5D.

At this junction, we would like to further explain the results of Figure 4 and Figure 5. The modification of ANTs is undetectable at 2h psi in samples using the wild-type Lp02 infection (Figure 4A and 4C), which is consistent with the results that WT infection doesn’t detectably affect the ADP/ATP exchange in mitochondria (Figure 5D). The strain that could detectably inhibit the ADP/ATP exchange activity of mitochondria is strain ∆*ceg3*(pCeg3), which overexpresses Ceg3 from a multicopy plasmid (Figure 5D). Infection by this strain allowed us to detect ADP-ribosylation of ANTs at the 2h infection time point (Figure 4A). Taken together, these results indicate that ANTs modification and ADP/ATP exchange inhibition occur concurrently, which suggests that the results in Figure 4 and Figure 5 are consistent.

References:

Rolando, M., Escoll, P., Nora, T., Botti, J., Boitez, V., Bedia, C., Daniels, C., Abraham, G., Stogios, P.J., Skarina, T., et al. (2016). Legionella pneumophila S1P-lyase targets host sphingolipid metabolism and restrains autophagy. Proc Natl Acad Sci U S A 113, 1901-1906.

Zhu, W., Hammad, L.A., Hsu, F., Mao, Y., and Luo, Z.Q. (2013). Induction of caspase 3 activation by multiple Legionella pneumophila Dot/Icm substrates. Cell Microbiol 15, 1783-1795.